# On Frank-Wolfe and Equilibrium Computation

**Jacob Abernethy**
Georgia Institute of Technology
`prof@gatech.edu`

**Jun-Kun Wang**
Georgia Institute of Technology
`jimwang@gatech.edu`

## Abstract

We consider the Frank-Wolfe (FW) method for constrained convex optimization, and we show that this classical technique can be interpreted from a different perspective: FW emerges as the computation of an equilibrium (saddle point) of a special convex-concave zero sum game. This saddle-point trick relies on the existence of no-regret online learning to both generate a sequence of iterates but also to provide a proof of convergence through vanishing regret. We show that our stated equivalence has several nice properties, as it exhibits a modularity that gives rise to various old and new algorithms. We explore a few such resulting methods, and provide experimental results to demonstrate correctness and efficiency.

## 1   Introduction

There has been a burst of interest in a technique known as the *Frank-Wolfe method* (FW) [10], also known as *conditional gradient*, for solving constrained optimization problems. FW is entirely a first-order method, does not require any projection operation, and instead relies on access to a linear optimization oracle. Given a compact and convex constraint set $X \subset \mathbb{R}^d$, we require the ability to (quickly) answer queries of the form $\mathcal{O}(v) := \arg\min_{x \in X} x^\top v$, for any vector $v \in \mathbb{R}^d$. Other techniques such as gradient descent methods require repeated projections into the constraint set which can be prohibitively expensive. Interior point algorithms, such as Newton path following schemes [1], require computing a hessian inverse at each iteration which generally does not scale well with the dimension.

In the present paper we aim to give a new perspective on the Frank-Wolfe method by showing that, in a broad sense, it can be viewed as a special case of *equilibrium computation via online learning*. Indeed, when the optimization objective is cast as a particular convex-concave payoff function, then we are able to extract the desired optimal point via the equilibrium of the associated zero-sum game. Within Machine Learning there has been a lot of attention paid to the computation of optimal strategies for zero-sum games using online learning techniques. An amazing result, attributed to [12] yet now practically folklore in the literature, says that we can compute the optimal equilibrium in a zero sum game by *pitting two online learning strategies against each other* and, as long as they achieve the desired regret-minimization guarantee, the long-run empirical average of their actions (strategy choices) must converge to the optimal equilibrium. This trick is both very beautiful but also extremely useful: it was in some sense the core of early work in Boosting [11], has been shown to generalize many linear programming techniques [3], it serves as the key tool for recent advances in flow optimization problems [8], and has been instrumental in understanding *differential privacy* [9].

We begin in Section 2 by reviewing the method of proving a generalized minimax theorem using regret minimization, and we show how this proof is actually constructive and gives rise to a generic meta-algorithm. This meta-algorithm is especially modular, and allows for the substitution of various algorithmic tools that achieve, up to convergence rates, essentially the same core result. We then show that the original Frank-Wolfe algorithm is simply one instantiation of this meta-algorithm, yet where the convergence rate follows as a trivial consequence of main theorem, albeit with an additional $O(\log T)$ factor.

We build upon this by showing that a number of variants of Frank-Wolfe are also simple instantiations of our meta-algorithm, with a convergence rate that follows easily. For example, we propose the *cumulative gradient* variant of Frank-Wolfe and prove that the same guarantee holds, yet relies on a potentially more stable optimization oracle. We show that techniques of [31] using stochastic smoothing corresponding to implement a Follow-the-perturbed-leader variant of our meta-algorithm. And finally, we use our framework to prove an entirely new result, showing that one obtains an $O(\log T/T)$ convergence rate even when the objective $f(\cdot)$ is not smooth, but instead the constraint set satisfies strong convexity.

The results laid out in this paper provide value not only in proving rates and establishing new and existing algorithms but also in setting forth a perspective on Frank-Wolfe-style methods that can leverage the wealth of results we have available from online learning and online convex optimization. At present, the possibilities and limits of various online learning problems has been thoroughly worked out [20, 7] with incredibly tight bounds. Using the connections we put forth, many of these results can provide a stronger theoretical framework towards understanding projection-free conditional gradient methods.

**Related works of projection-free algorithms**
[25] gives an analysis of FW for smooth objectives, and shows that FW converges at a $O(1/T)$ rate even when the linear oracle is solved approximately, under certain conditions. [30] develops a block-wise update strategy for FW on the dual objective of structural SVM, where only a subset of dual variables are updated at each iteration. In the algorithm, a smaller oracle is called due to the block-wise update, which reduces the computational time per iteration and leads to the speedup overall. [37] proposes updating multiple blocks at a time. [34] proposes using various measures to select a block for update.

In another direction, some results have aimed at obtaining improved convergence rates. [14] shows that for strongly convex and smooth objective functions, FW can achieve a $O(1/T^2)$ convergence rate over a strongly convex set. [13, 15] first show that one can achieve linear convergence for strongly convex and smooth objectives over polytopes using a projection-free algorithm. The algorithm constructs a stronger oracle which can be efficiently implemented for certain polytopes like simplex. [29] shows that some variants of FW such as away-step FW [38] or pairwise FW enjoy an exponential convergence rate when the feasible set is a polytope. [5] provides a refined analysis for the away-step FW. [17] extends [29] to some saddle-point optimization problems, where the constraint set is assumed to be a polytope and the objective is required to be strongly convex for one variable and strongly concave for the other. A drawback of away-step FW [38] is that it requires storing the previous outputs from the oracle. Very recently, [16] develop a new variant that avoids this issue for specific polytopes, which also enjoys exponential convergence for strongly convex and smooth objectives. Note that all of the exponential convergence results depend on some geometric properties of the underlying polytope.

Other works include variants for stochastic setting [23], online learning setting [22], minimizing some structural norms [19, 39], or reducing the number of gradient evaluations [32]. There is also a connection between subgradient descent and FW; Bach [4] shows that for certain types of objectives, subgradient descent applied to the primal domain is equivalent to FW applied to the dual domain.

**Preliminaries and Notation**

**Definition 1:** A convex set $Y \subseteq \mathbb{R}^m$ is an $\alpha$-*strongly convex set* w.r.t. a norm $\| \cdot \|$ if for any $u, v \in Y$, any $\theta \in [0, 1]$, the $\| \cdot \|$ ball centered at $\theta u + (1 - \theta)v$ with radius $\theta(1 - \theta)\frac{\alpha}{2}\|u - v\|^2$ is contained in $Y$. Please see [14] for examples about strongly-convex sets.

**Definition 2** A function is $\beta$-strongly smooth w.r.t. a norm $\| \cdot \|$ if $f$ is everywhere differentiable and $f(u) \leq f(v) + \nabla f(v)^\top (u - v) + \frac{\beta}{2}\|u - v\|^2$. A function is $\beta$-strongly convex w.r.t. a norm $\| \cdot \|$ if $f(u) \geq f(v) + \nabla f(v)^\top (u - v) + \frac{\beta}{2}\|u - v\|^2$.

**Definition 3** For a convex function $f(\cdot)$, its Fenchel conjugate is $f^*(x) := \sup_y \langle x, y \rangle - f(y)$. Note that if $f$ is convex then so is its conjugate $f^*$, since it is defined as the maximum over linear functions of $x$ [6]. Furthermore, the biconjugate $f^{**}$ equals $f$ if and only if $f$ is closed and convex. It is known that $f$ is $\beta$-strongly convex w.r.t. $\| \cdot \|$ if and only if $f^*$ is $1/\beta$ strongly smooth w.r.t the dual norm $\| \cdot \|_*$ [26], assuming that $f$ is a closed and convex function.

## 2 Minimax Duality via No-Regret Learning

### 2.1 Brief review of online learning

In the task of online convex optimization, we assume a *learner* is provided with a compact and convex set $K \subset \mathbb{R}^n$ known as the *decision set*. Then, in an online fashion, the learner is presented with a sequence of $T$ *loss functions* $\ell_1(\cdot), \ell_2(\cdot), \ldots, \ell_T(\cdot) : K \to \mathbb{R}$. On each round $t$, the learner must select a point $x_t \in K$, and is then "charged" a loss of $\ell_t(x_t)$ for this choice. Typically it is assumed that, when the learner selects $x_t$ on round $t$, she has observed all loss functions $\ell_1(\cdot), \ldots, \ell_{t-1}(\cdot)$ up to, but not including, time $t$. However, we will also consider learners that are *prescient*, i.e. that can choose $x_t$ with knowledge of the loss functions up to *and including* time $t$. The objective of interest in most of the online learning literature is the learner's *regret*, defined as $\mathcal{R}_T := \sum_{t=1}^{T} \ell_t(x_t) - \min_{x \in K} \sum_{t=1}^{T} \ell_t(x)$. Oftentimes we will want to refer to the *average regret*, or the regret normalized by the time horizon $T$, which we will call $\overline{\mathcal{R}}_T := \frac{\mathcal{R}_T}{T}$. What has become a cornerstone of online learning research has been the existence of *no-regret algorithms*, i.e. learning strategies that guarantee $\overline{\mathcal{R}}_T \to 0$ as $T \to \infty$.

Let us consider three very simple learning strategies, and we note the available guarantees for each.

(`FollowTheLeader`) Perhaps the most natural algorithm one might think of is to simply select $x_t$ as the *best point in hindsight*. That is, the learner can choose $x_t = \arg\min_{x \in K} \sum_{s=1}^{t-1} \ell_s(x)$.

**Lemma 1** ([21]). *If each $\ell_t(\cdot)$ is 1-lipschitz and 1-strongly convex, then `FollowTheLeader` achieves $\overline{\mathcal{R}}_T \le c\frac{\log T}{T}$ for some constant $c$.*

(`BeTheLeader`) When the learner is prescient, then we can do slightly better than `FollowTheLeader` by incorporating the current loss function: $x_t = \arg\min_{x \in K} \sum_{s=1}^{t} \ell_s(x)$. This algorithm was named `BeTheLeader` by [28], who also proved that it actually guarantees non-positive regret!

**Lemma 2** ([28]). *For any sequence of loss functions, `BeTheLeader` achieves $\overline{\mathcal{R}}_T \le 0$.*

(`BestResponse`) But perhaps the most trivial strategy for a prescient learner is to ignore the history of the $\ell_s$'s, and simply play the best choice of $x_t$ on the current round. We call this algorithm `BestResponse`, defined as $x_t = \arg\min_{x \in K} \ell_t(x)$. A quick inspection reveals that `BestResponse` satisfies $\overline{\mathcal{R}}_T \le 0$.

### 2.2 Minimax Duality

The celebrated *minimax theorem* for zero-sum games, first discovered by John von Neumann in the 1920s [36, 33], is certainly a foundational result in the theory of games. It states that two players, playing a game with zero-sum payoffs, each have an optimal randomized strategy that can be played obliviously – that is, even announcing their strategy in advance to an optimal opponent would not damage their own respective payoff, in expectation.

In this paper we will focus on more general minimax result, establishing duality for a class of convex/concave games, and we will show how this theorem can be proved without the need for Brouwer's Fixed Point Theorem [27]. The key inequality can be established through the use of no-regret learning strategies in online convex optimization, which we detail in the following section. The theorem below can be proved as well using Sion's Minimax Theorem [35].

**Theorem 1.** *Let $X, Y$ be compact convex subsets of $\mathbb{R}^n$ and $\mathbb{R}^m$ respectively. Let $g : X \times Y \to \mathbb{R}$ be convex in its first argument and concave in its second. Then we have that*

$$\min_{x \in X} \max_{y \in Y} g(x, y) = \max_{y \in Y} \min_{x \in X} g(x, y) \tag{1}$$

We want to emphasize that a meta-algorithm (Algorithm 1) actually emerges from our proof for Theorem 1, please see the supplementary for details. It is important to point out that the meta algorithm, as a routine for computing equlibria, is certainly not a novel technique, it has served implicitly as the underpinning of many works, including those already mentioned [11, 9, 8].

We close this section by summarizing the approximate equilibrium computation guarantee that follows from the above algorithm. This result is classical, and we explore it in great detail in the

---

**Algorithm 1** Meta Algorithm for equilibrium computation

---
1: Input: convex-concave payoff $g : X \times Y \to \mathbb{R}$, algorithms $\text{OAlg}^X$ and $\text{OAlg}^Y$
2: **for** $t = 1, 2, \ldots, T$ **do**
3: $\quad x_t := \text{OAlg}^X(g(\cdot, y_1), \ldots, g(\cdot, y_{t-1}))$
4: $\quad y_t := \text{OAlg}^Y(g(x_1, \cdot), \ldots, g(x_{t-1}, \cdot), g(x_t, \cdot))$
5: **end for**
6: Output: $\bar{x}_T = \frac{1}{T} \sum_{t=1}^T x_t$ and $\bar{y}_T := \frac{1}{T} \sum_{t=1}^T y_t$

---

**Appendix.** We let $\bar{x}_T := \frac{1}{T} \sum_{t=1}^T x_t$ and $\bar{y}_T := \frac{1}{T} \sum_{t=1}^T y_t$, and let $V^*$ be the value of the game, which is the quantity in (1).

**Theorem 2.** *Algorithm 1 outputs $\bar{x}_T$ and $\bar{y}_T$ satisfying*

$$\max_{y \in Y} g(\bar{x}_T, y) \le V^* + \epsilon_T + \delta_T \quad and \quad \min_{x \in X} g(x, \bar{y}_T) \ge V^* - (\epsilon_T + \delta_T). \qquad (2)$$

*as long as $OAlg^X$ and $OAlg^Y$ guarantee average regret bounded by $\epsilon_T$ and $\delta_T$, respectively.*

## 3    Relation to the Frank-Wolfe Method

We now return our attention to the problem of constrained optimization, and we review the standard Frank-Wolfe algorithm. We then use the technologies presented in the previous section to recast Frank-Wolfe as an equilibrium computation, and we show that indeed the vanilla algorithm is an instantiation of our meta-algorithm (Alg. 1). We then proceed to show that the modularity of the minimax duality perspective allows us to immediately reproduce existing variants of Frank-Wolfe, as well as construct new algorithms, with convergence rates provided immediately by Theorem 2.

To begin, let us assume that we have a compact set $Y \subset \mathbb{R}^n$ and a convex function $f : Y \to \mathbb{R}$. Our primary goal is to solve the objective

$$\min_{y \in Y} f(y). \qquad (3)$$

We say that $y_0$ is an $\epsilon$-*approximate* solution as long as $f(y_0) - \min_{y \in Y} f(y) \le \epsilon$.

### 3.1    A Brief Overview of Frank-Wolfe

---

**Algorithm 2** Standard Frank-Wolfe algorithm

---
1: **Input:** obj. $f : Y \to \mathbb{R}$, oracle $\mathcal{O}(\cdot)$, learning rate $\{\gamma_t \in [0,1]\}_{t=1,2,\ldots}$, init. $w_0 \in Y$
2: **for** $t = 1, 2, 3 \ldots, T$ **do**
3: $\quad v_t \leftarrow \mathcal{O}(\nabla f(w_{t-1})) = \arg\min_{v \in Y} \langle v, \nabla f(w_{t-1}) \rangle$
4: $\quad w_t \leftarrow (1 - \gamma_t) w_{t-1} + \gamma_t v_t.$
5: **end for**
6: Output: $w_T$

---

The standard Frank-Wolfe algorithm (Algorithm 2) consists of making repeated calls to a linear optimization oracle (line 6), followed by a convex averaging step of the current iterate and the oracle's output (line 7). It initializes a $w_1$ in the constraint set $Y$. Due to the convex combination step, the iterate $w_t$ is always within the constraint set, which is the reason why it is called projection free. We restate a proposition from [10], who established the convergence rate of their algorithm.

**Theorem 3** ([10]). *Assume that $f(\cdot)$ is 1-strongly smooth. If Algorithm 2 is run for $T$ rounds, then there exists a sequence $\{\gamma_t\}$ such that the output $w_T$ is a $O\left(\frac{1}{T}\right)$-approximate solution to (3).*

It is worth noting that the typical learning rate used throughout the literature is $\gamma_t = \frac{2}{2+t}$ [31, 25]. This emerges as the result of a recursive inequality.

## 3.2 Frank-Wolfe via the Meta-Algorithm

We now show that the meta-algorithm generalizes Frank-Wolfe, and provides a much more modular framework for producing similar algorithms. We will develop some of these novel methods and establish their convergence via Theorem 2.

In order to utilize minimax duality, we have to define decision sets for two players, and we must produce a convex-concave payoff function. First we will assume, for convenience, that $f(y) := \infty$ for any $y \notin Y$. That is, it takes the value $\infty$ outside of the convex/compact set $Y$, which ensures that $f$ is lower semi-continuous and convex. Now, let the $x$-player be given the set $X := \{\nabla f(y) : y \in Y\}$. One can check that the closure of the set $X$ is a convex set. Please see Appendix 2 for the proof.

**Theorem 4.** *The closure of (sub-)gradient space $\{\partial f(y) | y \in Y\}$ is a convex set.*

The $y$-player's decision set will be $Y$, the constraint set of the primary objective (3). The payoff $g(\cdot, \cdot)$ will be defined as

$$g(x, y) := -x^\top y + f^*(x). \tag{4}$$

The function $f^*(\cdot)$ is the Fenchel conjugate of $f$. We observe that $g(x, y)$ is indeed linear, and hence concave, in $y$, and it is also convex in $x$.

Let's notice a few things about this particular game. Looking at the $\max \min$ expression,

$$\max_{y \in Y} \min_{x \in X} g(x, y) = \max_{y \in Y} \left( - \max_{x \in X} \{x^\top y - f^*(x)\} \right) = - \left( \min_{y \in Y} f(y) \right) = V^*, \tag{5}$$

which follows by the fact that $f^{**} = f$.[1] Note, crucially, that the last term above corresponds to the objective we want to solve up to a minus sign. Any $\bar{y}$ which is an $\epsilon$-approximate equilibrium strategy for the $y$-player will also be an $\epsilon$-approximate solution to (3).

We now present the main result of this section, which is the connection between Frank-Wolfe (Alg. 2) and Alg. 1.

**Theorem 5.** *When both are run for exactly $T$ rounds, the output $\bar{y}_T$ of Algorithm 1 is identically the output $w_T$ of Algorithm 2 as long as: (I) Init. $x_1$ in Alg 1 equals $\nabla f(w_0)$ in Alg. 2; (II) Alg. 2 uses learning rate $\gamma_t := \frac{1}{t}$; (III) Alg. 1 receives $g(\cdot, \cdot)$ defined in (4); (IV) Alg. 1 sets $OAlg^X := \texttt{FollowTheLeader}$; (V) Alg. 1 sets $OAlg^Y := \texttt{BestResponse}$.*

*Proof.* We will prove that the following three equalities are maintained throughout both algorithms. We emphasize that the objects on the left correspond to Alg. 1 and those on the right to Alg. 2.

$$x_t = \nabla f(w_{t-1}) \tag{6}$$
$$y_t = v_t \tag{7}$$
$$\bar{y}_t = w_t. \tag{8}$$

We first note that the first condition of the theorem ensures that (6) holds for $t = 1$. Second, the choice of learning rate $\gamma_t = \frac{1}{t}$ already guarantees that (7) implies (8), since this choice of rate ensures that $w_t$ is always a uniform average of the updates $v_t$. It remains to establish (6) and (7) via induction. We begin with the former.

Recall that $x_t$ is selected via $\texttt{FollowTheLeader}$ against the sequence of loss functions $\ell_t(\cdot) := g(\cdot, y_t)$. To write precisely what this means,

$$
\begin{aligned}
x_t &:= \arg\min_{x \in X} \left\{ \frac{1}{t-1} \sum_{s=1}^{t-1} \ell_s(x) \right\} = \arg\min_{x \in X} \left\{ \frac{1}{t-1} \sum_{s=1}^{t-1} (-y_s^\top x + f^*(x)) \right\} \\
&= \arg\max_{x \in X} \left\{ \bar{y}_{t-1}^\top x - f^*(x) \right\} = \nabla f(\bar{y}_{t-1}).
\end{aligned}
$$

The final line follows as a result of the Legendre transform [6]. Of course, by induction, we have that $\bar{y}_{t-1} = w_{t-1}$, and hence we have established (6).

Finally, let us consider how $y_t$ is chosen according to `BestResponse`. Recall that sequence of loss functions presented to the $y$-player is $h_t(\cdot) := -g(x_t, \cdot)$. Utilizing `BestResponse` for this sequence implies that

$$y_t = \arg\min_{y \in Y} h_t(y) = \arg\min_{y \in Y} \left(x_t^\top y - f^*(x_t)\right) = \arg\min_{y \in Y} \left(x_t^\top y\right)$$

$$\text{((6) by induc.)} \quad = \arg\min_{y \in Y} \nabla f(\bar{y}_{t-1})^\top y \quad = \quad \arg\min_{y \in Y} \nabla f(w_{t-1})^\top y \quad (\text{ which is } v_t).$$

Where the last equality follows by induction via (8). This completes the proof. $\qquad\square$

Note that the algorithm does not need to compute the conjugate, $f^*$. While the Frank-Wolfe algorithm can be viewed as implicitly operating on the conjugate, it is only through the use of $\arg\max_{x \in X} \left\{\bar{y}_{t-1}^\top x - f^*(x)\right\}$. Yet, this operation does not need to be computed in the naive way (i.e. by first computing $f^*$ and then doing the maximization). Instead, the expression actually boils down to $\nabla f(y)$ which is just a gradient computation!

The equivalence we just established has several nice features. But it does not provide a convergence rate for Algorithm 2. This should perhaps not be surprising, as nowhere did we even use the smoothness of $f$ anywhere in the equivalence. Instead, this actually follows via a key application of Theorem 2, utilizing the fact that $f^*$ is strongly convex on the interior of the set $X$ [2], granting `FollowTheLeader` a logarithmic regret rate.

**Corollary 1.** *Assume that $f(\cdot)$ is 1-strongly smooth. Then Algorithm 2, with learning rate $\gamma_t := \frac{1}{t}$, outputs $w_T$ with approximation error $O\left(\frac{\log T}{T}\right)$.*

*Proof.* As a result of Theorem 5, we have established that Alg. 2 is a special case of Alg. 1, with the parameters laid out in the previous theorem. As a result of Theorem 2, the approximation error of $w_T$ is precisely the error $\epsilon_T + \delta_T$ of the point $\bar{y}_T$ when generated via Alg. 1 with subroutines $\text{OAlg}^X :=$ `FollowTheLeader` and $\text{OAlg}^Y = $ `BestResponse`, assuming that these two learning algorithms guarantee average regret no more than $\epsilon_T$ and $\delta_T$, respectively. We noted that `BestResponse` does not suffer regret, so $\delta_T = 0$.

To bound the regret of `FollowTheLeader` on the sequence of functions $g(\cdot, y_1), \ldots, g(\cdot, y_T)$, we observe that the smoothness of $f$ implies that $f^*$ is 1-strongly convex, which in turn implies that $g(x, y_t) = -x^\top y_t + f^*(x)$ is also 1-strongly convex (in $x$). Hence Lemma 1 guarantees that `FollowTheLeader` has average regret $\epsilon_T := O\left(\frac{\log T}{T}\right)$, which completes the proof. $\qquad\square$

We emphasize that the above result leans entirely on existing work on regret bounds for online learning, and these tools are doing the heavy lifting. We explore this further in the following section.

## 4 Frank-Wolfe-style Algs, New and Old

We now have a factory for generating new algorithms using the approach laid out in Section 3. Theorem 5 shows that the standard Frank-Wolfe algorithm (with a particular learning rate) is obtained via the meta-algorithm using two particular online learning algorithms $\text{OAlg}^X, \text{OAlg}^Y$. But we have full discretion to choose these two algorithms, as long as they provide the appropriate regret guarantees to ensure convergence.

### 4.1 Cumulative Gradients

We begin with one simple variant, which we call *Cumulative-Gradient Frank-Wolfe*, laid out in Algorithm 3. The one significant difference with vanilla Frank-Wolfe is that the linear optimization oracle receives as input the *average* of the gradients obtained thus far, as opposed to the last one.

**Algorithm 3** Cumulative-Gradient Frank-Wolfe

---
1: Initialize: any $w_0 \in Y$.
2: **for** $t = 1, 2, 3 \ldots, T$ **do**
3:     $v_t \leftarrow \arg\min_{v \in Y} \left\langle y, \frac{1}{t-1} \sum_{s=1}^{t-1} \nabla f(w_s) \right\rangle$
4:     $w_t \leftarrow (1 - \gamma_t) w_{t-1} + \gamma_t v_t$.
5: **end for**
6: Output: $w_T$

---

The proof of convergence requires little effort.

**Corollary 2.** *Assume that $f(\cdot)$ is 1-strongly smooth. Then Algorithm 3, with learning rate $\gamma_t := \frac{1}{t}$, outputs $w_T$ with approximation error $O\left(\frac{\log T}{T}\right)$.*

*Proof.* The result follows almost identically to Corollary 1. It requires a quick inspection to verify that the new linear optimization subroutine corresponds to implementing `BeTheLeader` as $\text{OAlg}^Y$ instead of `BestResponse`. However, both `BestResponse` and `BeTheLeader` have non-positive regret ($\delta_T \leq 0$) (Lemma 2 in the supplementary), and thus they achieve the same convergence.  □

We note that a similar algorithm to the above can be found in [31], although in their results they consider more general weighted averages over the gradients.

### 4.2 Perturbation Methods and Stochastic Smoothing

Looking carefully at the proof of Corollary 1, the fact that `FollowTheLeader` was suitable for the vanilla FW analysis relies heavily on the strong convexity of the functions $\ell_t(\cdot) := g(\cdot, y_t)$, which in turn results from the smoothness of $f(\cdot)$. But what about when $f(\cdot)$ is not smooth, is there an alternative algorithm available?

We observe that one of the nice techniques to grow out of the online learning community is the use of *perturbations* as a type of regularization to obtain vanishing regret guarantees [28] – their method is known as *Follow the Perturbed Leader* (FTPL). The main idea is to solve an optimization problem that has a random linear function added to the input, and to select[3] as $x_t$ the expectation of the $\arg\min$ under this perturbation. More precisely,

$$x_t := \mathbb{E}_Z \left[ \arg\min_{x \in X} \left\{ Z^\top x + \sum_{s=1}^{t-1} \ell_s(x) \right\} \right].$$

Here $Z$ is some random vector drawn according to an appropriately-chosen distribution and $\ell_s(x)$ is the loss function of the x-player on round $s$; with the definition of payoff function $g$, $\ell_s(x)$ is $-x^\top y_s + f^*(x)$ (4).

One can show that, as long as $Z$ is chosen from the right distribution, then this algorithm guarantees average regret on the order of $O\left(\frac{1}{\sqrt{T}}\right)$, although obtaining the correct dimension dependence relies on careful probabilistic analysis. Recent work of [2] shows that the analysis of perturbation-style algorithm reduces to curvature properties of a stochastically-smoothed Fenchel conjugate.

What is intriguing about this perturbation approach is that it ends up being equivalent to an existing method proposed by [31] (Section 3.3), who also uses a stochastically smoothed objective function. We note that

$$\mathbb{E}_Z \left[ \arg\min_{x \in X} \left\{ Z^\top x + \sum_{s=1}^{t-1} \ell_s(x) \right\} \right] = \mathbb{E}_Z \left[ \arg\max_{x \in X} \left\{ (\bar{y}_{t-1} + Z/(t-1))^\top x - f^*(x) \right\} \right]$$

$$= \mathbb{E}_Z [\nabla f(\bar{y}_{t-1} + Z/(t-1))] = \nabla \tilde{f}_{t-1}(\bar{y}_{t-1})$$

(9)

where $\tilde{f}_\alpha(x) := \mathbb{E}[f(x + Z/\alpha)]$. [31] suggests using precisely this modified $\tilde{f}$, and they prove a rate on the order of $O\left(\frac{1}{\sqrt{T}}\right)$. As discussed, the same would follow from vanishing regret of FTPL.

## 4.3 Boundary Frank-Wolfe

---
**Algorithm 4** Modified meta-algorithm, swapped roles

---
1: Input: convex-concave payoff $g : X \times Y \to \mathbb{R}$, algorithms OAlg$^X$ and OAlg$^Y$
2: **for** $t = 1, 2, \ldots, T$ **do**
3:      $y_t := \text{OAlg}^Y(g(x_1, \cdot), \ldots, g(x_{t-1}, \cdot))$
4:      $x_t := \text{OAlg}^X(g(\cdot, y_1), \ldots, g(\cdot, y_{t-1}), g(\cdot, y_t))$
5: **end for**
6: Output: $\bar{x}_T = \frac{1}{T} \sum_{t=1}^{T} x_t$ and $\bar{y}_T := \frac{1}{T} \sum_{t=1}^{T} y_t$

---

We observe that the meta-algorithm previously discussed assumed that the $x$-player was first to act, followed by the $y$-player who was allowed to be prescient. Here we reverse their roles, and we instead allow the $x$-player to be prescient. The new meta-algorithm is described in Algorithm 4. We are going to show that this framework lead to a new projection-free algorithm that works for non-smooth objective functions. Specifically, if the constraint set is strongly convex, then this exhibits a novel projection free algorithm that grants a $O(\log T/T)$ convergence even for non-smooth objective functions. The result relies on very recent work showing that `FollowTheLeader` for strongly convex sets [24] grants a $O(\log T)$ regret rate. Prior work has considered strongly convex decision sets [14], yet with the additional assumption that the objective is smooth and strongly convex, leading to $O(1/T^2)$ convergence. *Boundary Frank-Wolfe* requires neither smoothness nor strong convexity of the objective. What we have shown, essentially, is that a strongly convex boundary of the constraint set can be used in place of smoothness of $f(\cdot)$ in order to achieve $O(1/T)$ convergence.

---
**Algorithm 5** Boundary Frank-Wolfe

---
1: **Input:** objective $f : Y \to \mathbb{R}$, oracle $\mathcal{O}(\cdot)$ for $Y$, init. $y_1 \in Y$.
2: **for** $t = 2, 3 \ldots, T$ **do**
3:      $y_t \leftarrow \arg\min_{y \in Y} \frac{1}{t-1} \sum_{s=1}^{t-1} \langle y, \partial f(y_s) \rangle$
4: **end for**
5: Output: $\bar{y}_T = \frac{1}{T} \sum_{t=1}^{T} y_t$

---

We may now prove a result about Algorithm 5 using the same techniques laid out in Theorem 5.

**Theorem 6.** *Algorithm 5 is a instance of Algorithm 4 if **(I)** Init. $y_1$ in Alg 5 equals $y_1$ in Alg. 4; **(II)** Alg. 1 sets OAlg$^Y$ := `FollowTheLeader`; and **(III)** Alg. 1 sets OAlg$^X$ := `BestResponse`. Furthermore, when $Y$ is strongly convex, and $\sum_{s=1}^{t} \partial f(y_s)$ has non-zero norm, then*

$$f(\bar{y}_T) - \min_{y \in Y} f(y) = O(\frac{M \log T}{\alpha L_T T})$$

*where $M := \sup_{y \in Y} \|\partial f(y)\|$, $L_T := \min_{1 \le t \le T} \|\Theta_t\|$, $\Theta_t = \sum_{s=1}^{t} \frac{1}{t} \partial f(y_s)$.*

*Proof.* Please see Appendix 3 for the proof. □

Note that the rate depends crucially on $L_T$, which is the smallest averaged-gradient norm computed during the optimization. Depending on the underlying optimization problem, $L_T$ can be as small as $O(1/\sqrt{T})$ or can even be 0. Now let us discuss when the boundary FW works; namely, the condition that causes the cumulative gradient being nonzero. If a linear combination of gradients is $\mathbf{0}$ then clearly $\mathbf{0}$ is in the convex hull of subgradients $\partial f(x)$ for boundary points $x$. Since the closure of $\{\nabla f(x) | x \in Y\}$ is convex, according to Theorem 4, this implies that $\mathbf{0}$ is in $\{\nabla f(x) | x \in Y\}$. If we know in advance that $\mathbf{0} \notin \text{cl}(\{\nabla f(x) | x \in Y\})$ we are assured that the cumulative gradient will not be $\mathbf{0}$. Hence, the proposed algorithm may only be useful when it is known, a priori, that the solution $y^*$ will occur not in the interior but on the boundary of $Y$. It is indeed an odd condition, but it does hold in many typical scenarios. One may add a perturbed vector to the gradient and show that with high probability, $L_T$ is a non-zero number. The downside of this approach is that it would generally grant a slower convergence rate; it cannot achieve $\log(T)/T$ as the inclusion of the perturbation requires managing an additional trade-off.

## Footnotes

[1] It was important how we defined $X$ here, as the fenchel conjugate takes the value of $\infty$ at any point $x \notin \{\nabla f(y) : y \in Y\}$, hence the unconstrained supremum is the same as $\max_{x \in X}(\cdot)$

[2] We only need to assume $f$ is "smooth on the interior of $Y$" to get the result. (That $f$ is technically not smooth outside of Y is not particularly relevant) The result that $f^*$ is strongly convex on the interior of the set $X$ is essentially proven by [26] in their appendix. This argument has been made elsewhere in various forms in the literature (e.g. [18]).

[3]Technically speaking, the results of [28] only considered linear loss functions and hence their analysis did not require taking averages over the input perturbation. While we will not address computational issues here due to space, actually computing the average $\arg\min$ is indeed non-trivial.

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
