[Supplementary Material]

# On Frank-Wolfe and Equilibrium Computation (Supplementary)

**Jacob Abernethy**
Georgia Institute of Technology
`prof@gatech.edu`

**Jun-Kun Wang**
Georgia Institute of Technology
`jimwang@gatech.edu`

## 1    Minimax Duality via No-Regret Learning

The celebrated *minimax theorem* for zero-sum games, first discovered by John von Neumann in the 1920s [14, 10], is certainly a foundational result in the theory of games. It states that two players, playing a game with zero-sum payoffs, each have an optimal randomized strategy that can be played obliviously – that is, even announcing their strategy in advance to an optimal opponent would not damage their own respective payoff, in expectation. Or, if you are not fond of game theory, the statement can be stated quite simply in terms of the order of operations of a particular two-stage optimization problem: letting $\Delta_n$ and $\Delta_m$ be the $n-$ and $m-$probability simplex, respectively, and let $M \in \mathbb{R}^{n \times m}$ be any real-valued matrix, then we have

$$\min_{x \in \Delta_n} \max_{y \in \Delta_m} x^\top M y = \max_{y \in \Delta_m} \min_{x \in \Delta_n} x^\top M y.$$

One way to view this equality statement as the conjunction of two inequalities. A quick inspection tells us that $\min \max \geq \max \min$ is reasonably easy to prove in a sentence or two. But the other direction is non-trivial, and von Neumann's original proof rested on a deep result in topology, known as Brouwer's Fixed Point Theorem [6]. John Nash's famous proof of the existence of equilibria in general (non zero-sum) games was based on the same technology [9].

In this paper we will focus on a (simplified) generalization of the minimax result due to Maurice Sion [13], and indeed we will show how this theorem can be proved without the need for Brouwer. The key inequality can be established through the use of no-regret learning strategies in online convex optimization, which we detail in the following section. To begin, we state Sion's Minimax Theorem in a slightly restricted setting.[1]

**Theorem 1.** *Let $X, Y$ be compact convex subsets of $\mathbb{R}^n$ and $\mathbb{R}^m$ respectively. Let $g : X \times Y \to \mathbb{R}$ be convex in its first argument and concave in its second. Then we have that*

$$\min_{x \in X} \max_{y \in Y} g(x, y) = \max_{y \in Y} \min_{x \in X} g(x, y) \tag{1}$$

### 1.1    A Primer on No-Regret Algorithms

We now very briefly review the framework of online learning, online convex optimization, and no-regret algorithms. For a more thorough exposition, we refer the reader to the excellent surveys of [12], [3], and [1], among others.

In the task of online convex optimization, we assume a *learner* is provided with a compact and convex set $K \subset \mathbb{R}^n$ known as the *decision set*. Then, in an online fashion, the learner is presented with a sequence of $T$ loss functions $\ell_1(\cdot), \ell_2(\cdot), \ldots, \ell_T(\cdot) : K \to \mathbb{R}$. On each round $t$, the learner must select a point $x_t \in K$, and is then "charged" a loss of $\ell_t(x_t)$ for this choice. Typically it is assumed

that, when the learner selects $x_t$ on round $t$, she has observed all loss functions $\ell_1(\cdot), \ldots, \ell_{t-1}(\cdot)$ up to, but not including, time $t$. However, we will also consider learners that are *prescient*, i.e. that can choose $x_t$ with knowledge of the loss functions up to *and including* time $t$.

The objective of interest in most of the online learning literature is the learner's *regret*, defined as

$$\mathcal{R}_T := \sum_{t=1}^{T} \ell_t(x_t) - \min_{x \in K} \sum_{t=1}^{T} \ell_t(x).$$

Often times we will want to refer to the *average regret*, or the regret normalized by the time horizon $T$, which we will call $\overline{\mathcal{R}}_T := \frac{\mathcal{R}_T}{T}$. What has become a cornerstone of online learning research has been the existence of *no-regret algorithms*, i.e. learning strategies that guarantee $\overline{\mathcal{R}}_T \to 0$ as $T \to \infty$. And in many cases these algorithms are at the same time efficient and aesthetically pleasing.

Let us consider three very simple learning strategies[2], and we note the available guarantees for each.

(`FollowTheLeader`) Perhaps the most natural algorithm one might think of is to simply select $x_t$ as the *best point in hindsight*. That is, the learner can choose

$$x_t = \arg\min_{x \in K} \sum_{s=1}^{t-1} \ell_s(x).$$

This algorithm, coined `FollowTheLeader` by [7], does not provide vanishing regret in general. However, when every $\ell_t(\cdot)$ is *strongly convex* w.r.t. $L_2$, then we do obtain a non-trivial bound.

**Lemma 1** ([4])**.** *If each $\ell_t(\cdot)$ is 1-lipschitz and 1-strongly convex, then `FollowTheLeader` achieves $\overline{\mathcal{R}}_T \leq c \frac{\log T}{T}$ for some constant c.*

(`BeTheLeader`) When the learner is prescient, then we can do slightly better than `FollowTheLeader` by incorporating the current loss function:

$$x_t = \arg\min_{x \in K} \sum_{s=1}^{t} \ell_s(x).$$

This algorithm was named `BeTheLeader` by [7], who also proved that it actually guarantees non-positive regret!

**Lemma 2** ([7])**.** *For any sequence of loss functions, `BeTheLeader` achieves $\overline{\mathcal{R}}_T \leq 0$.*

*Proof.* We induct on $T$. The base case $T = 1$ is trivial. The inductive step proceeds as follows:

$$
\begin{array}{rcl}
\sum_{t=1}^{T} \ell_t(x_t) & = & \ell_T(x_T) + \sum_{t=1}^{T-1} \ell_t(x_t) \\
\text{(by induc.)} & \leq & \ell_T(x_T) + \min_{x \in K} \sum_{t=1}^{T-1} \ell_t(x) \\
\text{(substitute } x_T\text{)} & \leq & \ell_T(x_T) + \sum_{t=1}^{T-1} \ell_t(x_T) \\
\text{(def'n of } x_T\text{)} & = & \min_{x \in X} \sum_{t=1}^{T} \ell_t(x)
\end{array}
$$

$\square$

(`BestResponse`) But perhaps the most trivial strategy for a prescient learner is to ignore the history of the $\ell_s$'s, and simply play the best choice of $x_t$ on the current round. We call this algorithm `BestResponse`, defined as $x_t = \arg\min_{x \in K} \ell_t(x)$. A quick inspection reveals that `BestResponse` satisfies $\overline{\mathcal{R}}_T \leq 0$.

## 1.2 Minimax Proof via No Regret

Let us now return our attention to the proof of Theorem 1. We will utilize the tools laid out in the previous section.

First, we note that one direction of the proof is straightforward. The inequality $\min_x \max_y \geq \max_y \min_x$ follows by a simple inspection, by noting that a minimizing player would certain prefer to observe $x$ before selection $y$. We now proceed to the other direction, which is far less trivial.

In order to prove the second inequality, we are going to imagine that the $x$-player and the $y$-player are going to compete against each other, and each is going to choose their action according to some online learning strategy (to be determined later). Let us imagine the following setup: for $t = 1, 2, \ldots, T$ the $x$-learner chooses $x_t \in X$ and the $y$-learner selects $y_t \in Y$. From the perspective of the $x$-learner, on each round she is subject to a loss function $\ell_t(\cdot) : X \to \mathbb{R}$ defined as $\ell_t(\cdot) := g(\cdot, y_t)$. The $y$-learner, on the other hand, observes her own sequence of loss functions $h_t(\cdot) : Y \to \mathbb{R}$, defined as $h_t(\cdot) := -g(x_t, \cdot)$. Let us assume that both selected a learning algorithm that provides some guarantee on the average regret. That is, the $x$-player is assured that her average regret $\overline{\mathcal{R}}_T^x$ is upper bounded by $\epsilon_T$ for any sequence of $\ell_t$'s, and similarly the $y$-player has average regret $\overline{\mathcal{R}}_T^y \leq \delta_T$, for two sequences $\{\epsilon_T\}$ and $\{\delta_T\}$.

Let us now reason about the average cost to both players, $\frac{1}{T} \sum_{t=1}^{T} g(x_t, y_t)$, for playing according to their chosen learning strategies. First, we use the $y$-player's regret bound to obtain:

$$
\begin{aligned}
\frac{1}{T} \sum_{t=1}^{T} g(x_t, y_t) &= \frac{1}{T} \sum_{t=1}^{T} -h_t(y_t) \\
\text{(def. of } \overline{\mathcal{R}}_T^y) \quad &= -\min_{y \in Y} \left\{ \frac{1}{T} \sum_{t=1}^{T} h_t(y) \right\} - \overline{\mathcal{R}}_T^y \\
\text{(reg. bound)} \quad &\geq \max_{y \in Y} \left\{ \frac{1}{T} \sum_{t=1}^{T} g(x_t, y) \right\} - \delta_T \\
\text{(Jensen)} \quad &\geq \max_{y \in Y} g\left( \tfrac{1}{T} \sum_{t=1}^{T} x_t, y \right) - \delta_T \qquad (2) \\
&\geq \min_{x \in X} \max_{y \in Y} g(x, y) - \delta_T
\end{aligned}
$$

Let us now apply the same argument on the right hand side, where we use the $x$-player's regret guarantee.

$$
\begin{aligned}
\frac{1}{T} \sum_{t=1}^{T} g(x_t, y_t) &= \frac{1}{T} \sum_{t=1}^{T} \ell_t(x_t) \\
&= \min_{x \in X} \left\{ \frac{1}{T} \sum_{t=1}^{T} \ell_t(x) \right\} + \overline{\mathcal{R}}_T^x \\
&\leq \min_{x \in X} \left\{ \frac{1}{T} \sum_{t=1}^{T} g(x, y_t) \right\} + \epsilon_T \\
&\leq \min_{x \in X} g\left( x, \tfrac{1}{T} \sum_{t=1}^{T} y_t \right) + \epsilon_T \qquad (3) \\
&\leq \max_{y \in Y} \min_{x \in X} g(x, y) + \epsilon_T
\end{aligned}
$$

If we combine the two inequalities

$$
\min_{x \in X} \max_{y \in Y} g(x, y) - \delta_T \leq \max_{y \in Y} \min_{x \in X} g(x, y) + \epsilon_T
$$

However, as long as each player selected a no-regret strategy, the quantities $\epsilon_T$ and $\delta_T$ can be driven arbitrarily close to 0. We may then conclude that

$$
\min_{x \in X} \max_{y \in Y} g(x, y) \leq \max_{y \in Y} \min_{x \in X} g(x, y)
$$

and we are done.

## 2   Proof of Theorem 4

*Proof.* This is a result of the following lemmas.

**Definition:** *[Definition 12.1 in [11]] A mapping $T : \mathcal{R}^n \to \mathcal{R}^n$ is called monotone if it has the property that*

$$\langle v_1 - v_0, x_1 - x_0 \rangle \geq 0 \text{ whenever } v_0 \in T(x_0), v_1 \in T(x_1).$$

*Moreover, $T$ is maximal monotone if there is no monotone operator that properly contains it.*

**Lemma 2:** *[Theorem 12.17 in [11]] For a proper, lsc, convex function $f$, $\partial f$ is a maximum monotone operator.*

**Lemma 3:** *[Theorem 12.41 in [11]] For any maximal monotone mapping $T$, the set "domain of $T$" is nearly convex, in the sense that there is a convex set $C$ such that $C \subset$ domain of $T \subset cl(C)$. The same applies to the range of $T$.*

Therefore, the closure of $\{\partial f(y) | y \in Y)\}$ is also convex, because we can define another proper, lsc, convex function $\hat{f}(y)$ such that it is $\hat{f}(y) = f(y)$ if $y \in Y$; otherwise, $\hat{f}(y) = \infty$. Then, the sub-differential of $\hat{f}(y)$ is equal to $\{\partial f(y) | y \in Y\}$. So, we can apply the the lemmas to get the result. $\qquad \square$

## 3   Proof of Theorem 6

*Proof.* The proof builds on the latest result of FTL for linear losses on strongly convex sets [5]. Recall that FTL plays

$$y_t = \arg\min_{y \in Y} \sum_{s=1}^{t-1} \langle y, \ell_s \rangle = \arg\min_{y \in Y} \sum_{s=1}^{t-1} \langle y, -\Theta_{t-1} \rangle = \arg\max_{y \in Y} \langle y, \Theta_{t-1} \rangle$$

where $\Theta_{t-1} = -\sum_{s=1}^{t-1} \ell_s$. The analysis requires the solution obtained from FTL to be unique, which requires that the cumulative loss $\Theta_{t-1}$ bounded away from $0$ for each $t$. Let us denote the support function $\Phi(\cdot) \equiv \max_{y \in Y} \langle y, \cdot \rangle$. The following theorem is what the boundary FW relies on.

**Theorem 2.** *(Theorem 3.3 in [5]) Let $Y \subset \mathbb{R}^d$ be an $\alpha$-strongly convex set. Let $M = \max_s \|\ell_s\|$ and assume that $\Phi(\cdot)$ has a unique maximizer for each $\{\Theta_t\}_{\forall t}$. Define $L_T := \min_{1 \leq t \leq T} \|\Theta_t\|$. Choose $y_1 \in bd(Y)$. Then, after $T$ rounds, the regret $R_T = \frac{2M^2}{\alpha L_T}(1 + \log(T))$.*

Equipped with the theorem, we can know proof the new result. Note that the y-player is facing a sequence of loss functions $\psi_t(\cdot) := g(x_t, \cdot) = \langle -x_t, \cdot \rangle + f^*(x_t)$ in each round $t$. Since the y-player plays FTL,

$$y_t := \arg\max_{y \in Y} \frac{1}{t-1} \sum_{s=1}^{t-1} \psi_s(y) = \arg\max_{y \in Y} \frac{1}{t-1} \sum_{s=1}^{t-1} -x_s^\top y + f^*(x_s) = \arg\max_{y \in Y} \frac{1}{t-1} \sum_{s=1}^{t-1} -x_s^\top y. \tag{4}$$

According to Theorem 2,

$$\max_{y \in Y} \sum_{t=1}^{T} \psi_t(y) - \sum_{t=1}^{T} \psi_t(y_t) \leq O\left(\frac{M \log T}{\alpha L_T}\right), \tag{5}$$

where $M$ here represents the upper bound of the gradient norm, as we will see soon.

So, (5) is equivalent to

$$\frac{1}{T} \max_{y \in Y} \sum_{t=1}^{T} g(x_t, y) - \frac{1}{T} \sum_{t=1}^{T} g(x_t, y_t) \leq \delta_T, \tag{6}$$

where $\delta_T = O(\frac{M \log T}{\alpha L_T T})$. This give us the following result

$$
\frac{1}{T} \sum_{t=1}^{T} g(x_t, y_t) \geq \frac{1}{T} \max_{y \in Y} \sum_{t=1}^{T} g(x_t, y) - \delta_T = \max_{y \in Y} \frac{1}{T} \sum_{t=1}^{T} g(x_t, y) - \delta_T
$$

$$
\geq \max_{y \in Y} g(\sum_{t=1}^{T} \frac{1}{T} x_t, y) - \delta_T \geq \min_{x \in X} \max_{y \in Y} g(x, y) - \delta_T. \tag{7}
$$

Now for the x-player, since it plays `BestResponse`

$$
x_t := \arg \min_{x \in X} g(x, y_t) = \arg \min_{x \in X} \{-y_t^\top x + f^*(x)\} = \arg \max_{x \in X} \{y_t^\top x - f^*(x)\} = \partial f(y_t). \tag{8}
$$

Then,

$$
\frac{1}{T} \sum_{t=1}^{T} g(x_t, y_t) = \frac{1}{T} \sum_{t=1}^{T} \min_{x \in X} g(x, y_t) \leq \frac{1}{T} \sum_{t=1}^{T} \max_{y \in Y} \min_{x \in X} g(x, y) = \max_{y \in Y} \min_{x \in X} g(x, y) \tag{9}
$$

Combining (7) and (9), we get

$$
\min_{x \in X} \max_{y \in Y} g(x, y) \leq \max_{y \in Y} \min_{x \in X} g(x, y) + \delta_T.
$$

Moreover, from (7) and using the fact that $V^* = \max_{y \in Y} \min_{x \in X} g(x, y) = \min_{x \in X} \max_{y \in Y} g(x, y)$, we have

$$
V^* - \delta_T \leq \frac{1}{T} \sum_{t=1}^{T} g(x_t, y_t) = \frac{1}{T} \sum_{t=1}^{T} \min_{x \in X} g(x, y_t). \tag{10}
$$

We know that $V^* = -\min_{y \in Y} f(y)$ by (5). Moreover, $\min_{x \in X} g(x, y_t) = -f(y_t)$. Combining the results and (10), we get

$$
\frac{1}{T} \sum_{t=1}^{T} f(y_t) - \min_{y \in Y} f(y) \leq \delta_T. \tag{11}
$$

Let $\bar{y}_T = \frac{1}{T} \sum_{t=1}^{T} y_t$. We now obtain the result by Jensen's inequality.

$$
f(\bar{y}_T) - \min_{y \in Y} f(y) \leq \delta_T = O(\frac{M \log T}{\alpha L_T T}). \tag{12}
$$

$\square$

## 4 Experiments

To justify the theoretical result, we implemented Algorithm 5 on real data. Our objective was to solve a hinge-loss minimization problem over the L2 ball. The latter is indeed a strongly convex set.

$$
\min_{w: \|w\|_2 \leq k} \sum_{i=1}^{N} \max(0, 1 - l_i w^\top z_i), \tag{13}
$$

where $(z_i, l_i)$ are the feature vector and label for sample $i$ and $w$ are the parameters to be learned.

The objective function is a non-smooth function. We compare the boundary FW (Algorithm 5) with the smoothing algorithm in [8] which we've described in Section 4.2 and here we call it "smoothing FW". The original smoothing FW in [8] requires uniformly sampling points in the unit ball. For implementation simplicity, we use Gaussian perturbation instead, which has the same effect in smoothing a non-smooth function [2]. We conduct the experiment on three datasets: a9a, mushroom, and phishing [3].

The figures show that boundary FW has a better convergence rate than smoothing FW. We also note that the iteration cost of smoothing FW is much more expensive than boundary FW due to its sampling and smoothing step.

Figure 1: The objective vs. number of iteration. For boundary FW, we plot $f(\bar{y}_t)$.

# 5  Recorvering the standard Frank-Wolfe with the common step size

We recall that the step size of standard Frank-Wofle is $\gamma_t = \frac{t}{t+2}$. Indeed, let us show that the two-player zero-sum game framework can actually capture the FW with standard step size as well. This time the loss functions faced by both players is modified through scaling by a multiplicative factor. Specifically, in each round $t$, the loss function of the x-player is $\alpha_t \ell_t(\cdot) : X \to \mathbb{R}$, where $\ell_t(\cdot) := g(\cdot, y_t)$. The y-player, on the other hand, observes her own sequence of loss functions $\alpha_t h_t(\cdot) : Y \to \mathbb{R}$, where $h_t(\cdot) := -g(x_t, \cdot)$. Suppose there are $T$ rounds, we requre that $\Sigma_{t=1}^{T} \alpha_t = T$, where each $\alpha_t \geq 0$ and each $\alpha_t$ would be revealed when a learner receives the loss function in round $t$. Considering the y-player,

$$
\begin{aligned}
\frac{1}{T}\sum_{t=1}^{T}\alpha_t g(x_t, y_t) &= \frac{1}{T}\sum_{t=1}^{T} -\alpha_t h_t(y_t) \\
(\text{def. of } \overline{\mathcal{R}}_T^y) &= -\frac{1}{T}\min_{y\in Y}\left\{\sum_{t=1}^{T}\alpha_t h_t(y)\right\} - \overline{\mathcal{R}}_T^y \\
(\text{reg. bound}) &\geq \max_{y\in Y}\left\{\frac{1}{T}\sum_{t=1}^{T}\alpha_t g(x_t, y)\right\} - \delta_T \\
(\text{Jensen}) &\geq \max_{y\in Y} g\left(\frac{1}{T}\sum_{t=1}^{T}\alpha_t x_t, y\right) - \delta_T \qquad (14)\\
&\geq \min_{x\in X}\max_{y\in Y} g(x, y) - \delta_T
\end{aligned}
$$

Let us now apply the same argument using the $x$-player's regret guarantee.

$$
\begin{aligned}
\frac{1}{T}\sum_{t=1}^{T}\alpha_t g(x_t, y_t) &= \frac{1}{T}\sum_{t=1}^{T}\alpha_t \ell_t(x_t) \\
&= \min_{x\in X}\left\{\sum_{t=1}^{T}\frac{1}{T}\alpha_t \ell_t(x)\right\} + \overline{\mathcal{R}}_T^x \\
&\leq \min_{x\in X}\left\{\sum_{t=1}^{T}\frac{1}{T}\alpha_t g(x, y_t)\right\} + \epsilon_T \\
&\leq \min_{x\in X} g\left(x, \sum_{t=1}^{T}\frac{1}{T}\alpha_t y_t\right) + \epsilon_T \qquad (15)\\
&\leq \max_{y\in Y}\min_{x\in X} g(x, y) + \epsilon_T
\end{aligned}
$$

**Theorem 3.** $\frac{1}{T}\sum_{t=1}^{T}\alpha_t x_t$ *and* $\frac{1}{T}\sum_{t=1}^{T}\alpha_t y_t$ *satisfying*

$$
\max_{y\in Y} g\left(\frac{1}{T}\sum_{t=1}^{T}\alpha_t x_t, y\right) \leq V^* + \epsilon_T + \delta_T \quad \text{and} \quad \min_{x\in X} g\left(x, \frac{1}{T}\sum_{t=1}^{T}\alpha_t y_t\right) \geq V^* - (\epsilon_T + \delta_T). \quad (16)
$$

*as long as $OAlg^X$ and $OAlg^Y$ guarantee average regret bounded by $\epsilon_T$ and $\delta_T$, respectively.*

We remind the readers of the standard Frank-Wolfe algorithm, with step size $\gamma_t = \frac{2}{t+2}$, described in Algorithm 1.

---

**Algorithm 1** Standard Frank-Wolfe algorithm

---

1: **Input:** obj. $f : Y \to \mathbb{R}$, oracle $\mathcal{O}(\cdot)$, learning rate $\gamma_t = \frac{2}{t+2}$, init. $w_0 \in Y$
2: **for** $t = 0, 1, 2, 3 \dots, T$ **do**
3:     $v_t \leftarrow \mathcal{O}(\nabla f(w_t)) = \arg\min_{v \in Y} \langle v, \nabla f(w_t) \rangle$
4:     $w_{t+1} \leftarrow (1 - \gamma_t) w_t + \gamma_t v_t$.
5: **end for**
6: Output: $w_T$

---

We want to show the following meta-algorithm captures the standard Frank-Wofle with step size $\gamma_t = \frac{2}{t+2}$.

---

**Algorithm 2** Meta Algorithm for equilibrium computation

---

1: Input: convex-concave payoff $g : X \times Y \to \mathbb{R}$, algorithms $OAlg^X$ and $OAlg^Y$,
2: Require: $\Sigma_{t=1}^T \alpha_t = T$ and each $\alpha_t \geq 0$.
3: **for** $t = 1, 2, \dots, T$ **do**
4:     $x_t := OAlg^X(\alpha_1 g(\cdot, y_1), \dots, \alpha_{t-1} g(\cdot, y_{t-1}))$
5:     $y_t := OAlg^Y(\alpha_1 g(x_1, \cdot), \dots, \alpha_{t-1} g(x_{t-1}, \cdot), \alpha_t g(x_t, \cdot))$
6: **end for**
7: Output: $\bar{x}_T := \frac{1}{T} \sum_{t=1}^T \alpha_t x_t$ and $\bar{y}_T := \frac{1}{T} \sum_{t=1}^T \alpha_t y_t$

---

As before, we define

$$g(x, y) := -x^\top y + f^*(x). \tag{17}$$

But, in each round, the loss function the $x$-learner receives is $\alpha_t \ell_t(\cdot) := \alpha_t g(\cdot, y_t)$. The $y$-learner, on the other hand, observes her loss functions $\alpha_t h_t(\cdot) := -\alpha_t g(x_t, \cdot)$.

**Theorem 4.** *Let $\alpha_1 : \alpha_2 : \cdots : \alpha_t : \cdots : \alpha_T = 1 : 2 : \cdots : t : \cdots : T$ and that $\Sigma_{t=1}^T \alpha_t = T$ and $\alpha_t \geq 0, \forall t$.*

*When both are run for exactly $T$ rounds, the output $\bar{y}_T := \frac{1}{T} \sum_{t=1}^T \alpha_t y_t$ of Algorithm 2 is identically the output $w_T$ of Algorithm 1 as long as: (I) Init. $x_1$ in Alg 2 equals $\nabla f(w_0)$ in Alg. 1; (II) Alg. 1 uses learning rate $\gamma_t := \frac{2}{t+2}$; (III) Alg. 2 receives $g(\cdot, \cdot)$ defined in (17); (IV) Alg. 2 sets $OAlg^X := FollowTheLeader$ (V) Alg. 2 sets $OAlg^Y := BestResponse$.*

*Proof.* One can show that in the standard Frank-Wolfe with step size $\gamma_t = \frac{2}{t+2}$,

$$
\begin{aligned}
w_1 &= \gamma_0 v_1 = v_1 \\
w_2 &= (1 - \gamma_1)\gamma_0 v_1 + \gamma_1 v_2 = \frac{1}{3} v_1 + \frac{2}{3} v_2 \\
w_3 &= (1 - \gamma_2)(1 - \gamma_1)\gamma_0 v_1 + (1 - \gamma_2)\gamma_1 v_2 + \gamma_2 v_3 = \frac{1}{6} v_1 + \frac{2}{6} v_2 + \frac{3}{6} v_3 \\
w_4 &= \frac{3}{30} v_1 + \frac{6}{30} v_2 + \frac{9}{30} v_3 + \frac{12}{30} v_4 \\
&\cdots
\end{aligned}
\tag{18}
$$

We want to show that

$$
\begin{aligned}
x_t &= \nabla f(w_{t-1}) \\
y_t &= v_t
\end{aligned}
\tag{19}
$$
$$\tag{20}$$

$$
\sum_{s=1}^t \frac{\alpha_s}{\left( \sum_{s=1}^t \alpha_s \right)} y_s = w_t \tag{21}
$$

Recall that $x_t$ is selected via `FollowTheLeader` against the sequence of loss functions $\alpha_t \ell_t(\cdot) := \alpha_t g(\cdot, y_t)$,

$$
\begin{aligned}
x_t &:= \arg\min_{x \in X} \left\{ \frac{1}{t-1} \sum_{s=1}^{t-1} \alpha_s \ell_s(x) \right\} \\
&= \arg\min_{x \in X} \left\{ \frac{1}{t-1} \sum_{s=1}^{t-1} \alpha_s (-y_s^\top x + f^*(x)) \right\} \\
&= \arg\min_{x \in X} \left\{ \sum_{s=1}^{t-1} \alpha_s (-y_s^\top x + f^*(x)) \right\} \\
&= \arg\max_{x \in X} \left\{ \sum_{s=1}^{t-1} \alpha_s y_s^\top x - (\sum_{s=1}^{t-1} \alpha_s) f^*(x)) \right\} \\
&= \arg\max_{x \in X} \left\{ \sum_{s=1}^{t-1} \frac{\alpha_s}{(\sum_{s=1}^{t-1} \alpha_s)} y_s^\top x - f^*(x) \right\}
\end{aligned}
\tag{22}
$$

Given the sequence $\{\alpha_s\}$ satisfies $\alpha_1 : \alpha_2 : \cdots : \alpha_{t-1} = 1 : 2 : \cdots : t-1$, it follows that $x_t = \nabla f(\sum_{s=1}^{t-1} \frac{\alpha_s}{(\sum_{s=1}^{t-1} \alpha_s)} y_s) = \nabla f(w_{t-1})$, since $\sum_{s=1}^{t-1} \frac{\alpha_s}{(\sum_{s=1}^{t-1} \alpha_s)} y_s = \sum_{s=1}^{t-1} \frac{\alpha_s}{(\sum_{s=1}^{t-1} \alpha_s)} v_s = w_{t-1}$.

Finally,

$$
\begin{aligned}
y_t &= \arg\min_{y \in Y} \alpha_t h_t(y) = \arg\min_{y \in Y} \alpha_t \left( x_t^\top y - f^*(x_t) \right) = \arg\min_{y \in Y} \alpha_t \left( x_t^\top y \right) \\
&= \arg\min_{y \in Y} \nabla f(\sum_{s=1}^{t-1} \frac{\alpha_s}{(\sum_{s=1}^{t-1} \alpha_s)} y_s)^\top y \quad = \quad \arg\min_{y \in Y} \nabla f(w_{t-1})^\top y \quad (\text{ which is } v_t).
\end{aligned}
$$

$\square$

## Footnotes

[1]In Sion's fully general version, no finite-dimensional assumptions were required, the set $Y$ need not be compact, and $g(x, \cdot)$ need only be quasi-concave and upper semicontinuous on $Y$, and $g(\cdot, y)$ quasi-convex and lower semicontinuous on $X$.

[2]We observe that many algorithms we describe do not have a well-defined approach to choosing the initial point $x_1$, in which case $x_1$ can be selected arbitrarily from the decision set. Also, whenever we specify $x_t = \arg\min(\ldots)$ and the solution set is non-unique then any minimizer is acceptable.

[3] Available on `https://www.csie.ntu.edu.tw/~cjlin/libsvmtools/datasets/`