[Reviews · NeurIPS 2017]

Reviewer 1



The paper draws a connection between the classical Frank-Wolfe algorithm for constrained smooth & convex optimization (aka conditional gradient method) and using online learning algorithms to solve zero-sum games. This connection is made by casting the constrained convex optimization problem as a convex-concave saddle point problem between a player that takes actions in the feasible set and another player that takes actions in the gradient space of the objective function. This saddle point problem is derived using the Flenchel conjugate of the objective. Once this is achieved, a known and well explored paradigm of using online learning algorithms can be applied to solving this saddle point problem (where each player applies its own online algorithm to either minimize or maximize), and the average regret bounds obtained by the algorithms translate back to the approximation error with respect to the objective on the original offline convex optimization problem. The authors show that by applying this paradigam with different kinds of online learning algorithms, they can recover the original Frank-Wolfe algorithm (though with a slightly different step size and rate worse by a factor of log(T)) and several other variants, including one that uses the averaged gradient, using stochastic smoothing for non-smooth objectives and even a new variant that converges for non-smooth objectives (without smoothing), when the feasible set is strongly convex. ***Criticism*** Pros: I think this is a really nice paper, and I enjoyed reading it. While, in hindsight, the connection between offline optimization and zero-sum games via Fenchel conjugates is not surprising, I find this perspective very refreshing and original, in particular in light of the connection with online learning. I think NIPS is the best place for works making such a connection between offline optimizaiton and online learning. I truly believe this might open a window to obtain new and improved projection-free algorithms via the lens of online learning, and this is the main and worthy contribution of this paper, in my opinion. This view already allows the authors to give a FW-like algorithm for non-smooth convex optimization over strongly convex sets which overcomes the difficulties of the original method. There is an easy example of minimizing a piece-wise linear function over the euclidean unit ball for which FW fails, but this new variant works (the trick is to indeed use the averaged gradient). Unfortunately, as I discuss next, the authors do a somewhat sloppy work presenting this result. Cons: - I find Section 4.3 to be very misleading. Algorithm 5 is a first-order method for convex non-smooth optimization. We know that in this setting the best possible (worst-case) rate is 1/\eps^2. The construction is simply minimization of a piece-wise linear function over a unit Euclidean ball (which is certainly strongly convex with a constant strong convexity parameter). Hence, claiming Algorithm 5 obtains a log(T)/T rate is simply not true and completely misleading. Indeed, when examining the details of the above bad" construction, we have that L_T will be of order \eps when aiming for \eps error. The authors should do a better job in correctly placing the result in the context of offline optimization, and not simply doing plug&play from the online world. Again, stating the rate as log(T)/T is simply misleading. In the same vain, I find the claims on lines 288-289 to be completely false. It is also disappointing and in a sense unsatisfactory that the algorithm completely fails when gradient=0 is achievable for the problem at hand. I do hope the authors will be able to squeeze more out of this very nice result (as I said, it overcomes a standard lower bound for original FW I had in mind). Minor things: - When referring to previous works [13,15] the authors wrongly claim that they rely on a stronger oracle. They also require the standard linear oracle but use it to construct a stronger oracle. Hence, they work for every polytope. - In Definition 2 is better to state it with gradient instead of subgradient since we assume the function is smooth. -Line 12: replace an with a - on several occasions when using ...=argmin", note in general the argmin is a set, not a singleton. Same goes for the subdifferential. -Line 103: I believe often times" should be a single word - Theorem 3: should state that there exists a choice for the sequence \gamma_t such that... - Theorem 5: should also state the gamma_t in the corresponding Algorithm 2 - Supp material Definition on line 82: centered equation is missing >=0. Also, properly contains" in line 84 not defined. - Supp material: Section 4 is missing (though would love to hear the details (: ) *** post rebuttal *** I've read the rebuttal and overall satisfied with the authors response.

Reviewer 2



This paper looks at how Frank Wolfe algorithm can be cast as a minimax optimization problem which can be cast as an equilibrium computation of a zero-sum game, where two players both play optimally. The paper shows that the standard Frank Wolfe method can be cast as an example of a minimax game, where the first player plays the Follow the Leader algorithm while the second player plays Best Response algorithm. They show that the regret of various online algorithms including FTL in a smooth setting can be obtained from the analysis of this framework. The same sort of analysis for the second player playing Best Leader gives an analysis of cumulative conjugate gradient algorithm analysis. The meta-algorithm and Frank wolfe algorithm can also be extended to the setting when f is not smooth, so as to match the FTPL algorithm. In this case, the two players are switched with the second player playing FTL and the first player playing Best Response. which ends up providing the Boundary Frank Wolfe algorithm with O(1/T) rates of convergence with a strongly convex boundary. The paper is well written and is easy to read with the connection shown nicely in a mathematical format. However it is not very clear how many novel results fall under the purview of the paper's method. The connection between FW and the minimax optimization is pretty well known [17] in the paper's references talk about it in detail. It would be interesting to know what allows the analyses to work for the non-smooth case, also it would be interesting to get some perspective if any of the analyses can match the best rates of batch methods as well.